# Structural and Electrical Investigation of Cobalt-Doped NiO*_x_*/Perovskite Interface for Efficient Inverted Solar Cells

**DOI:** 10.3390/nano10050872

**Published:** 2020-04-30

**Authors:** Zahra Rezay Marand, Ahmad Kermanpur, Fathallah Karimzadeh, Eva M. Barea, Ehsan Hassanabadi, Elham Halvani Anaraki, Beatriz Julián-López, Sofia Masi, Iván Mora-Seró

**Affiliations:** 1Institute of Advanced Materials (INAM), Universitat Jaume I, Av. Sos Baynat, s/n, 12071 Castelló, Spain; z.rezay@ma.iut.ac.ir (Z.R.M.); barea@uji.es (E.M.B.); hassanab@uji.es (E.H.); julian@uji.es (B.J.-L.); 2Department of Materials Engineering, Isfahan University of Technology, Isfahan 84156-83111, Iran; ahmad_k@cc.iut.ac.ir (A.K.); karimzadeh_f@cc.iut.ac.ir (F.K.); elhamhalvani@gmail.com (E.H.A.); 3Textile Engineering Department, Textile Excellence & Research Centers, Amirkabir University of Technology, Tehran 15916-34311, Iran

**Keywords:** inverted planar perovskite solar cell, hole transport material, Co-doped NiO*_x_*, perovskite morphology, electrical conductivity

## Abstract

Inorganic hole-transporting materials (HTMs) for stable and cheap inverted perovskite-based solar cells are highly desired. In this context, NiO*_x_*, with low synthesis temperature, has been employed. However, the low conductivity and the large number of defects limit the boost of the efficiency. An approach to improve the conductivity is metal doping. In this work, we have synthesized cobalt-doped NiO*_x_* nanoparticles containing 0.75, 1, 1.25, 2.5, and 5 mol% cobalt (Co) ions to be used for the inverted planar perovskite solar cells. The best efficiency of the devices utilizing the low temperature-deposited Co-doped NiO*_x_* HTM obtained a champion photoconversion efficiency of 16.42%, with 0.75 mol% of doping. Interestingly, we demonstrated that the improvement is not from an increase of the conductivity of the NiO*_x_* film, but due to the improvement of the perovskite layer morphology. We observe that the Co-doping raises the interfacial recombination of the device but more importantly improves the perovskite morphology, enlarging grain size and reducing the density of bulk defects and the bulk recombination. In the case of 0.75 mol% of doping, the beneficial effects do not just compensate for the deleterious one but increase performance further. Therefore, 0.75 mol% Co doping results in a significant improvement in the performance of NiO*_x_*-based inverted planar perovskite solar cells, and represents a good compromise to synthesize, and deposit, the inorganic material at low temperature, without losing the performance, due to the strong impact on the structural properties of the perovskite. This work highlights the importance of the interface from two different points of view, electrical and structural, recognizing the role of a low doping Co concentration, as a key to improve the inverted perovskite-based solar cells’ performance.

## 1. Introduction

Organic-inorganic halide perovskite solar cells (PSCs) have gained increasing attention owing to their high power conversion efficiencies and fabrication in solution at low temperatures. Within just a few years, PSCs have boosted efficiency to 25.2% [1]. Moreover, low-temperature processing makes them promising for future industrialization [2]. Along with the improvement in the perovskite material and deposition techniques, the advances in perovskite solar cells are also due to the study of the different hole-transporting and electron-transporting materials (HTMs and ETMs, respectively) employed in the devices [3]. The two most common architecture used in the fabrication of PSCs are the *n*-*i*-*p* and *p*-*i*-*n* configurations and the current highest efficiencies are achieved with the *n*-*i*-*p* one, using TiO_2_ as ETM and doped Spiro-OMeTAD as HTM [4]. This configuration is characterized by the high cost of the HTM and by a slight hysteresis, which has to be avoided with the use of special treatment or interlayers [5]. This hysteretic behavior can also be suppressed by using inverted *p*-*i*-*n* architectures [6,7,8]. Moreover, the planar *p*-*i*-*n* configuration is more favorable due to its relatively low-temperature processing, primarily when organic HTM is used [9,10]. However, one of the challenges in the fabrication of inverted perovskite solar cells is the lack of stable and low-cost HTM: i.e., poly (3, 4-ethylen edioxythiophene):poly (styrenesulfonate) (PEDOT:PSS) has high acidity, which decreases the long-term stability of the devices [11], and the poly[*N*,*N’*-bis(4-butylphenyl)-*N*,*N’*-bisphenylbenzidine] (p-TPD) and the poly(triaryl amine) (PTAA) are expensive and very hydrophobic, hindering the easy perovskite deposition, due to the poor wettability [12]. Conversely, inorganic materials like, MoO*_x_*, CuCrO_2_, CuO*_x_*, CuSbS_2_, CuSCN and NiO*_x_*, have low-cost and superior stability as well as higher mobility, basic prerequisites for obtaining stable and high-efficiency perovskite solar cells.

Among the inorganic HTMs, the low-temperature NiO_x_ is the most common in PSCs [13,14,15,16]. The remarkable properties of NiO*_x_*, such as intrinsic p-type doping nature, high optical transmittance, deep-lying valence band (VB) (5.4 eV) and low cost, make NiO*_x_* a preferential candidate for HTM in PSCs. NiO*_x_* has been prepared by several methods such as spray pyrolysis and a solution process or pulsed laser deposition [14,17,18,19]. Despite the merits of NiO*_x_* as HTM, its low intrinsic electrical conductivity and the high amount of defects when it is synthetized at low temperature, resulting in an increased charge recombination and reduced hole extraction [20], have blocked further improvement to NiO*_x_*-based PSCs. To overcome this drawback, keeping the temperature low, two main methods have been investigated, one is the addition of interlayers to control the interface [21] and a more effective one is metal ion doping. Different metals are used to dope the NiO*_x_*, like Au^+^ [13], Cu^2+^ [22], Co^3+^ [23] and Li^+^ [24]. Among different metals, Co is a promising dopant, yet used for other materials in direct solar cells configuration [25,26,27]. With regard to the NiO*_x_*, due to the low lattice parameter mismatch of 1.6% [28], a power conversion efficiency (PCE) of 18% has been obtained [29], but with an annealing around 300–400 °C [30]. Low-temperature synthesis of Co-doped NiO*_x_* has been further optimized, and an improvement of the conductivity when the level of doping increases has been demonstrated [17]. However, the maximum efficiency is obtained at a doping level (1%) lower than that needed to obtain the maximum conductivity (5%), because a decrease of the hole extraction ability [17]. Here we have developed the low temperature synthesis of Co-doped NiO*_x_* nanoparticles with Co ion from 0.75 to 5 mol% to stay in the range of higher conductivity and good hole extraction properties [23]. A comparative analysis of the structural and optical properties of the perovskite material, when deposited on the inorganic HTM, is also carried out. Although the conductivity is almost the same in the case of the NiO*_x_* doped and not doped by Co, we demonstrated that the Co doping leads to an increase of perovskite layer morphology and a change in the recombination mechanism. This is correlated with the increased fill factor (FF) in the solar cells—usually low due to the high temperature deposition (Table 1) employed and for the generation of defects in the bare NiO*_x_* devices—in turn due to the increased perovskite crystallinity and to the morphological improvement when the Co doping is employed as high as 0.75 mol%, assisting in the reduction of the more hostile bulk recombination [31,32,33].

## 2. Experimental Section

### 2.1. Materials

Ni(NO_3_)_2_·6H_2_O (99.9%) and CoCl_2_·6H_2_O (99.0%), were purchased from Sigma Aldrich (Madrid, Spain). Lead iodide (PbI_2_, >98%, from TCI, Tokyo, Japan), methylammonium iodide (MAI, 98%, from Greatcellsolar, Queanbeyan, Australia), 2-propanol (99.7% from Carlo Erba, Val de Reuil, France), ethanol (96%) and acetone (99.25%) from PanReac (Castellar del Valles, Spain), hydrochloric acid (HCl 37%), dimethyl sulfoxide (DMSO anhydrous 99.9%), chlorobenzene (CB anhydrous 99.8%), ethylacetate (EA anhydrous 99.8%), zinc powder (99.995%) from Sigma aldrich. [6,6]-Phenyl-C61-butyric acid methyl ester (PCBM, 99%) was purchased from nano-C (Westwood, NJ, USA). 2,9-Dimethyl-4,7-diphenyl-1,10-phenanthroline (BCP, 99%) was purchased from Sigma Aldrich.

### 2.2. Synthesis of NiO_x_ and Co-Doped NiO_x_ Nanoparticles

The NiO*_x_* was synthetized following previous reports [23]. The different steps of the synthesis are summarized in Figure 1. NiO*_x_* nanoparticles doped with different amounts of Co with different molar ratios (*x* = 0, 0.75, 1, 1.25, 2.5 and 5 mol%) were synthesized using the chemical co-precipitation method, according to the method reported in literature [35]. The typical procedure is as follows: Ni(NO_3_)_2_·6H_2_O (0.25 mol) was dissolved in 50 mL of deionized deoxygenated H_2_O to obtain a dark green solution. After being vigorously magnetically stirred for 1 h, NaOH solution (10 M, alkali source) in drops was added to the initial solution by slower stirring for crystallizing structures. During this operation, the initial dark green color of the solution turned to opaque pale green as a result of nanoparticles formation. Then keeping it stirred for another 1 h, the pale green colloidal precipitate was collected by an ultrasonic centrifugation at 6500 rpm for 15 min, and washed twice with deionized water. Finally, the product was dried at 80 °C overnight. The green solid obtained was grinded in mortar then calcined at 270 °C for 2 h to obtain a black powder. It is worth mentioning that all main synthesis steps were performed under N_2_ gas protection to create an oxygen-free atmosphere and prevent oxidation of divalent nickel salts. For doping, divalent Co (Co^2+^), CoCl_2_·6H_2_O was added to the Ni(NO_3_)_2_·6H_2_O solution at various molar ratios which were mentioned above. Before use, nanoparticles were dispersed in deionized water by ultrasonics probe for 2 min with amplitudes of 52% and cycle 1 (20 mg mL^−1^). Finally, the nanoparticle solution was filtered through a PVDF filter (0.40 μm).

### 2.3. Fabrication of Solar Cells

Substrates (tin-doped indium oxide-ITO) were etched with zinc powder and HCl 6 M. After cleaning with soap and water and then placing in an ultrasonic bath with 2% Extran solution in water for 15 min with ethanol, isopropanol, and acetone for 15 min for each solvent, the substrate were dried by airflow and were put in an ultraviolet–ozone (UV-O_3_) environment for 10 min to remove organic residues. The HTMs were obtained by spin-coating the corresponding NiO*_x_* or Co-NiO*_x_* aqueous solution (20 mg mL^−1^) with a speed of 3000 rpm for 40 s and heated at 130 °C for 10 min, in ambient conditions. The perovskite film (MAPbI_3_) was deposited over the HTM layer by one-step spin coating at 2000 rpm for 10 s, followed by 6000 rpm for 28 s with an acceleration of 3000, with the precursor composed of PbI_2_ and MAI (1.3 M for each) in 1 mL DMSO. Ethyl acetate was dropped onto the perovskite film at the last 15th second during the spin coating. As soon as the spin coating was finished, the sample was moved to a hotplate and annealed for 10 min at a temperature 130 °C [36]. Afterward, PCBM (20 mg mL^−1^ in CB) was spin-coated at 1000 rpm for 30 s and dried at 60 °C for 10 min. BCP (0.5 mg mL^−1^ in 2-propanol) was deposited by spin-coating at 6000 rpm for 30 s and dried at 60 °C for 5 min. Finally, Ag electrodes with a thickness of 100 nm were thermally evaporated at a deposition rate of about 0.5 nm s^−1^ in a vacuum chamber through a shadow mask [37].

### 2.4. Structural Characterization

The morphologies of the samples (ITO/Co-NiO*_x_*/MAPbI_3_) were characterized with a field-emission scanning electron microscope (FEG-SEM JEOL 3100F) operated at 5 kV, equipped with energy-dispersive X-ray spectroscopy. The X-ray diffraction (XRD) patterns of the NiO*_x_* and Co-NiO*_x_* powders were recorded using an X-ray diffractometer (D8 Advance, Bruker AXS, Karlsruhe, Germany) (Cu Kα, the wavelength of λ = 1.5406 Å) within the range of 30–70°. Transmission electron microscopy (TEM) images were recorded using (JEOL 2100 microscope, Akishima, Japan). Surface morphologies of thin films were observed using an atomic force microscope (AFM) (CSI-Nano Observer, Les Ulis, France).

### 2.5. Optoelectronic Characterization

The current−voltage (*J*/*V*) curves were measured using a Keithley 2612 source meter under AM 1.5 G (100 mWcm^−2^) provided by a solar simulator model 69,920 Newport. Each curve was generated using 123 data points. The active area of the cell is 0.121 cm^2^, and the scan rate was 10 mVs^−1^. The incident photons to current efficiency (IPCE) measurements were performed with a QEPVSI-b Oriel measurement system. The steady-state absorption spectra of the perovskite films were achieved by using an ultraviolet–visible (UV/Vis) absorption spectrophotometer (Varian, Cary 300, Palo Alto, CA, USA), and the steady state and the time-resolved photoluminescence (PL) decay were collected by a Horiba Fluorolog. The steady state PL was collected at a wavelength of 780 nm after excitation at 532 nm.

## 3. Results and Discussion

To investigate in detail how the presence of very low content of Co dopant (0.75–5 mol%) in the nickel oxide structure influencing the properties of the PSCs, a first characterization of NiO*_x_* nanoparticles was made. TEM images of NiO*_x_* nanoparticles and NiO*_x_* with different Co percentage as dopant are shown in Figure 2. Pure NiO*_x_* nanoparticles have a broad size distribution (10–33 nm) centered at 17 nm. However, the Co-doped particles present a smaller average size around 12 nm, and a narrower size distribution than the undoped nanoparticles. These results are in good agreement with the crystal size calculated from the XRD measurements (Figure 3a) using the Debye–Scherrer formula, see Appendix A [38,39].

XRD peaks of all samples correspond to the standard pattern characteristic of the cubic crystal structure of NiO*_x_*, with three characteristic diffraction peaks at 37.1°, 43.0° and 62.7° related to (111), (200) and (220) planes of NiO*_x_*, respectively (Figure 3a). No secondary phase is detected, ruling out the formation of other Co-based structures such as CoO. Therefore, the NiO*_x_* has been formed with relatively high phase purity and, according to the difference in the peaks intensities between the samples of undoped NiO*_x_* and Co-doped NiO*_x_*, it is clear that Ni ions have been successfully replaced with Co ions, as demonstrated also by energy-dispersive X-ray spectroscopy (EDS) measurements (Appendix A) [40]. We investigated the optical characteristics of the pristine NiO*_x_* and Co-NiO*_x_* films. The Co-NiO*_x_* films deposited on the ITO substrates revealed a high transmission (>87%) in the visible region (400–800 nm) (Figure 3b). [23] Concerning the optical properties of the Co-NiO*_x_* films, they showed a slightly lower transmission than undoped films, see Figure 3b.

NiO*_x_* and Co-doped NiO*_x_* nanoparticles were deposited in thin film in order to work as HTM. The surface morphologies of pure NiO*_x_* and Co-NiO*_x_* films are shown in Figure 4. The Co-NiO*_x_* films consisted of smaller-sized grains than the pure NiO*_x_* film and the 0.75 mol% show the most uniform morphology over a large area [23].

MAPbI_3_ films were deposited on top of the NiO*_x_* or Co-NiO*_x_* HTM. The SEM images (Figure 5a–f) indicated similar polycrystalline morphologies, uniform and pinhole-free perovskite with smaller grain sizes (∼150–200 nm) on the top of NiO*_x_* film and the biggest size is for the Co doping percentage 2.5 and 5 mol%). The bigger grain size could be due to better morphology of the Co-doped substrate. Between the different doping percentages, the 0.75 mol% Co-NiO*_x_* film, that present the best morphology, produce also the second for dimension perovskite grains, of ∼250–300 nm (Figure 5b), together with a more uniform distribution (Appendix A). The uniformity of the inorganic HTM and the big grains of the perovskite, demonstrate the potential of the 0.75 mol% doping for the fabrication of high-quality optoelectronic devices. The XRD spectra of perovskite layers (Figure 5g) show the presence of six main peaks at 14.20°, 19.94°, 23.56°, 28.50°, 31.75°, and 40.67°, which correspond to the (110) (112) (202) (220) (310) and (224) planes for the perovskite MAPI_3_ [36]. Moreover, the perovskite growth on the NiO*_x_* substrate doped with 0.75 mol% of Co is the more crystalline and oriented one, as the intensity of the peak at 14.20° and the ratio between the (110) and (220) faces are the highest (Appendix A). The benefits of growing crystal with large grain size and good orientation are, firstly, the reduced grain boundary area associated with large grains. The reduced grain boundary area suppresses bulk defects, charge trapping and decreases recombination, with the consequence of a relative higher carrier mobility. This is also translated in the device, in which the photo-generated carriers can easily propagate without frequent encounters with defects and impurities [41], see below.

To quantify the effect of Co-NiO*_x_* as HTMs, PSCs have been fabricated and characterized with an inverted structure as shown in Figure 6a (glass/ITO/NiO*_x_* or Co-NiO*_x_*/MAPbI_3_/PCBM/BCP/Ag). The cross section (Figure 6b,c) demonstrates that the thickness of the perovskite is the same (300 nm) on the top of bare NiO*_x_* and Co-NiO*_x_*. It is noted here that better charge transport and low energy loss could be expected for the devices with Co-NiO*_x_* because of the well-matched energy levels to perovskite than the pristine NiO*_x_* [34]. The pristine champion NiO*_x_*-based inverted PSC showed a fill factor (FF) of 70%, a short-circuit current density (*J_sc_*) of 19.5 mA cm^−2^, a *V_oc_* of 0.96 V and a final PCE of 13.2%. By the insertion of the Co ions, *J_sc_*, FF and *V_oc_* increase, compared with that of pristine NiO*_x_* (Figure 6d) [42,43]. The corresponding device parameters are summarized in Table 2 and the statistics on 40 devices for each type are reported in Appendix A. We achieved a champion PCE of 16.42%, for the 0.75 mol% Co-NiO*_x_*, close to the PCE reported with NiO*_x_* as HTM synthetized at high temperature [29] and the highest reported for low temperature Co-doped NiO*_x_* based inverted perovskite solar cells [23]. The highest FF value was found again for the 0.75 mol% Co-NiO*_x_* device with a value of 76%, compared to 70% for the pristine NiO*_x_* device. Thus, it is clear that a proper Co doping effectively improves the quality of the perovskite materials and enhances the *J_sc_* and the general device performance. The stabilized current at maximum power point is also reported in Appendix A. Note that despite 5 mol% Co-NiO*_x_* samples present big grain sizes, see Figure 5f, they also present a low shut resistance, see Figure 6d, that decreases significantly the FF, see Table 2. This results points to the presence of pinholes probably induced by the fast growth of some grains.

The IPCE spectra of the champion device shows higher external quantum efficiency for 0.75 mol% Co-NiO*_x_* than the prinstine NiO_x_ device (Figure 6e). The integrated current densities derived from the IPCE spectrum are 21.5 mAcm^−2^ for 0.75 mol% Co-NiO*_x_* and 19.5 mAcm^−2^ for prinstine NiO*_x_*, in good agreement with the *J*/*V* measurements. Moreover, the *J_sc_* obtained is the highest reported for low-temperature Co-doped NiO_x_ [23] not caused to the different active layer thickness (Figure 6b,c).

In order to unveil the origin of the better performance of Co-NiO*_x_* samples a systematic optical and electrical characterization was performed. The behavior of the photoluminescence (PL) with NiO*_x_* and Co-NiO*_x_* HTM has been analyzed. Comparing the PL of MAPbI_3_ films deposited on glass with the PL of MAPbI_3_ deposited on NiO*_x_* and Co-NiO*_x_* films, a significant PL quenching is observed, especially for samples deposited on Co-NiO*_x_* films, see Figure 7a and Appendix A. This fact indicates an increase of the non-radiative recombination at the interface [44]. This point is partially confirmed by the decrease of the PL lifetime measured by time-resolved PL (TRPL), see Figure 7b and Appendix A. The perovskite layer without any HTM showed the average lifetime (*τ_ave_*) of photo-generated excitons of 32 ns, while the perovskite layer on pristine NiO*_x_* and Co-NiO*_x_* exhibited *τ_ave_* of 20 ns and 9.8 ns (Figure 7b).

On the other hand, surface roughness and electrical properties of the HTM were analyzed by atomic force microscopy (AFM) and conductive AFM (c-AFM). At this doping level, the improvement of the conductivity is not significant, as we measure by conductive-AFM (Appendix A). The surface roughness of NiO*_x_* and Co-NiO*_x_* films is also the same at about 50 nm, see Appendix A. Since the films roughness was still comparable to the pristine NiO*_x_* film, it confirms again that the Co was effectively embedded in the NiO*_x_* without notable deformation of the rock salt crystal structure [34]. Meanwhile c-AFM measurement shows similar electrical behavior (Appendix A), thus the improvement of performances of PSCs based on Co-NiO*_x_* cannot be attributed to an increase of the electrical conductivity. Therefore, an analysis of the ideality factor, *m*, of both devices was performed to characterize whether the effect of Co in the structure, in addition to improving the crystallinity and grain size, is to modify the electrical behavior of the device, specifically the electronic recombination.

In order to analyze the recombination, the mechanism *m* of both samples was obtained from the slope of the *V_oc_* dependence with light illumination, see Figure 7c, using the relation: [45,46].
(1)$$e\cdot V_{OC}=E_{g}+m\cdot k_{B}\cdot T\cdot ln\frac{\phi }{\phi_{0}}$$
where *e* is the electron charge, *E_g_* is the light absorber bandgap, *k_B_* is the Boltzmann constant, *T* the temperature, *Φ* is the light intensity and *Φ_0_* is a constant with the same units than *Φ*.

For the Co-NiO_x_ device, two different behaviors are observed, corresponding to high and low light intensity. We determined *m* = 4.2 at low light intensities corresponding to a bulk multiple trapping in trap distribution. However, at high light irradiation *m* decreases to 1.5, indicating that most of traps are filled at higher light intensity and that the recombination mechanism is now mainly influenced by surface recombination, in good agreement with the PL results. For the NiO*_x_* device (Figure 7c) the recombination mechanism is totally independent from the light intensity, as the device presents the same ideal factor of 4.8, pointing to a higher density of bulk traps, which cannot be filled even at high light intensity.

Taking into account the different characterizations performed on the various devices using NiO*_x_* and Co-NiO*_x_*, focusing concretely on 0.75 mol% that presents the highest performance, we observed two different main trends. On the one hand, an increase of the grain size and improvement of the morphology, which decrease the bulk defects, and on the other hand an increase of the surface recombination. In the case of Co-NiO*_x_* 0.75 mol%, it presents higher and more uniform grain size and we hypothesize that the positive effect of the improvement of the bulk properties compensate for the deleterious effect of a not-suppressed interfacial recombination, causing a final enhancement of the device performance.

## 4. Conclusions

In summary, we have investigated the suitability of Co-doped NiO*_x_* nanoparticles as HTMs. Co-NiO*_x_* nanoparticles with a diameter of ∼10–20 nm, containing different concentrations (0–5 mol%) of Co ions were synthesized. A comprehensive analysis of optical, morphological and crystallographic investigations demonstrated a double effect of the Co doping. On the one hand, Co doping arises from a surface recombination, on the other hand, Co doping influences the properties of the perovskite grown on top of the Co-NiO*_x_* substrates, improving the morphology as a consequence of a grain size enlargement, and reducing the bulk-recombination. The final performance of the devices prepared with Co-NiO*_x_* as HTM depends on the balance between these two opposite trends. The case of 0.75 mol% Co-NiO*_x_*-based device, that presents a perovskite active layer with big grain size and the highest crystallinity, produces a positive balance and the devices with the highest performance are obtained with champion PCE of 16.42%, higher than previous results with the Co-NiO*_x_* HTM synthetized at low temperature.

## Figures and Tables

**Figure 1 nanomaterials-10-00872-f001:**
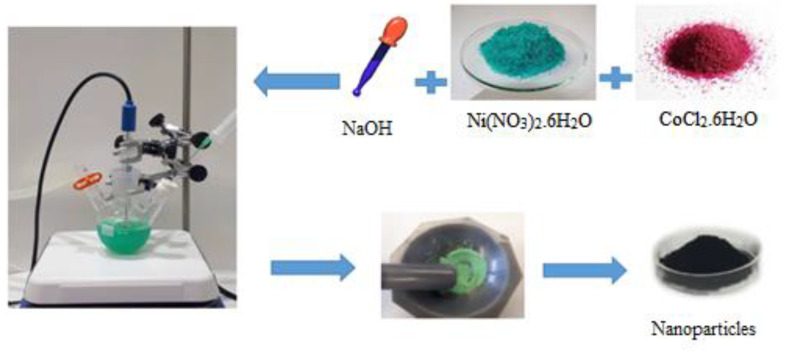
Sketch of the Co-doped NiO*_x_* nanoparticles synthesis.

**Figure 2 nanomaterials-10-00872-f002:**
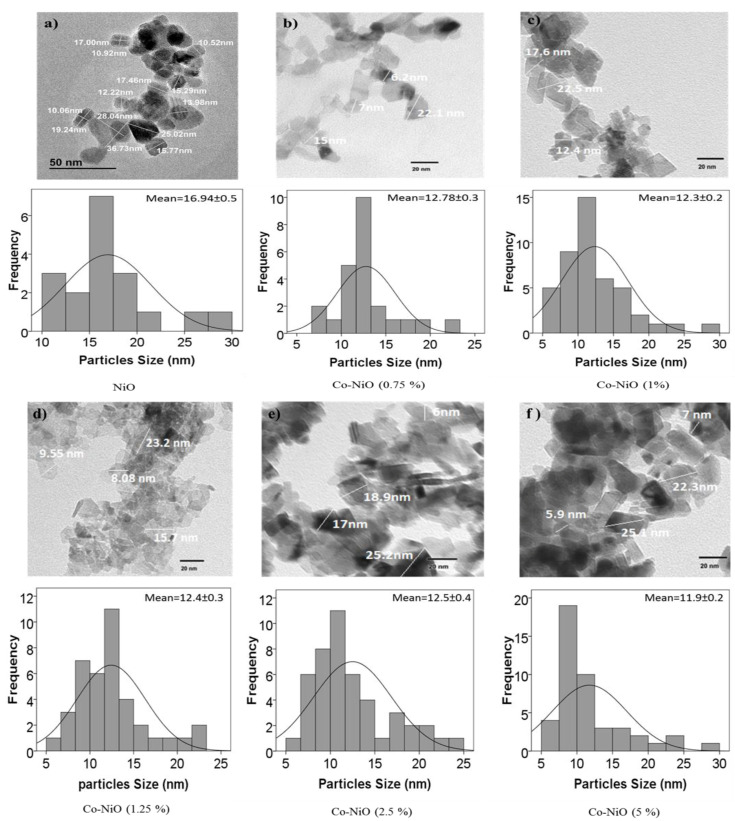
Transmission electron microscopy (TEM) images and relative histograms of the nanoparticles size distributions of (**a**) NiO*_x_* nanoparticles; (**b**) 0.75 mol% Co-NiO*_x_* nanoparticles; (**c**) 1 mol% Co-NiO*_x_* nanoparticles; (**d**) 1.25 mol% Co-NiO*_x_* nanoparticles; (**e**) 2.5 mol% Co-NiO*_x_* nanoparticles; (**f**) 5 mol% Co-NiO*_x_* nanoparticles.

**Figure 3 nanomaterials-10-00872-f003:**
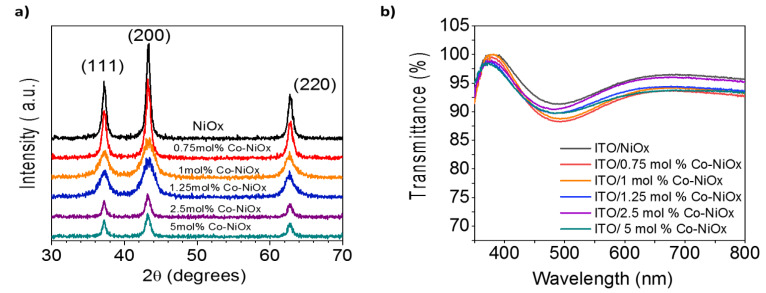
(**a**) X-ray diffraction (XRD) patterns of the NiO*_x_* and Co-doped NiO*_x_* nanoparticles; (**b**) optical transmission spectra for the NiO*_x_* and Co-NiO*_x_* films fabricated on top of ITO substrates, where the thickness of the Co-NiO*_x_* films was around 37 nm.

**Figure 4 nanomaterials-10-00872-f004:**
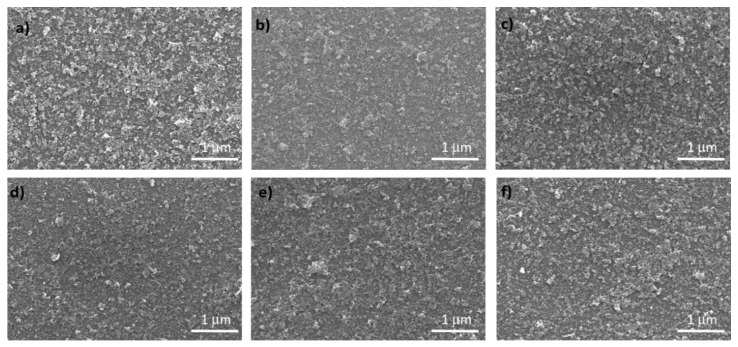
Scanning electron microscopy (SEM) images of the NiO*_x_* and Co-NiO*_x_* films on glass substrate. (**a**) NiO*_x_*; (**b**) 0.75 mol% Co-NiO*_x_*; (**c**) 1 mol% Co-NiO*_x_*; (**d**) 1.25 mol% Co-NiO*_x_*; (**e**) 2.5 mol% Co-NiO*_x_*; (**f**) 5 mol% Co-NiO*_x_*.

**Figure 5 nanomaterials-10-00872-f005:**
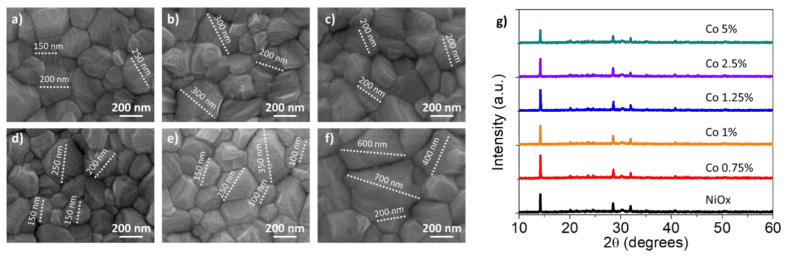
Top view SEM images for perovskite layers fabricated on the top of NiO*_x_* or Co-NiO*_x_* layer; (**a**) NiO*_x_*. (**b**) 0.75 mol% Co-NiO*_x_*; (**c**) 1 mol% Co-NiO*_x_*; (**d**) 1.25 mol% Co-NiO*_x_*; (**e**) 2.5 mol% Co-NiO*_x_*; (**f**) 5 mol% Co-NiO*_x_*; (**g**) corresponding perovskite X-ray diffraction patterns.

**Figure 6 nanomaterials-10-00872-f006:**
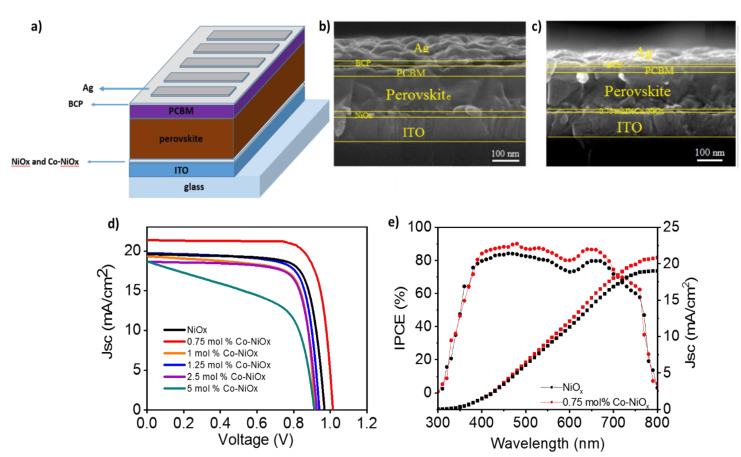
(**a**) Sketch of the device architecture; Cross-sectional SEM images for perovskite layers fabricated on the top of NiO*_x_* or Co-NiO*_x_* layer: (**b**) NiO*_x_* and (**c**) 0.75 mol% Co-NiO*_x_*; (**d**) *J*/*V* curves of PSCs using 0.75–5 mol% Co-NiO*_x_* and bare NiO*_x_* as hole-transporter materials (HTMs). (**e**) incident photons to current efficiency (IPCE) of 0.75 mol% Co-NiO*_x_* and NiO*_x_* based champion base devices.

**Figure 7 nanomaterials-10-00872-f007:**
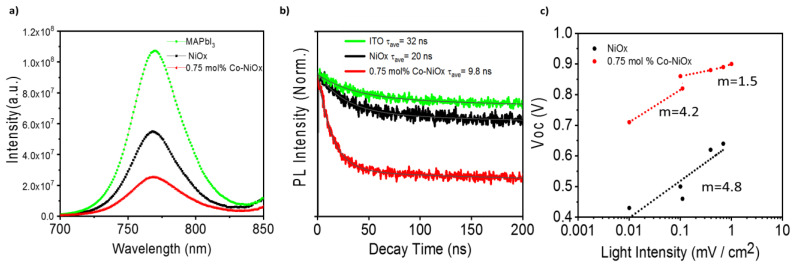
(**a**) Photoluminescence (PL) of MAPbI_3_ films on top of NiO*_x_* and 0.75 mol% Co-NiO*_x_*. The MAPbI_3_ on top of glass was used as a reference measurement; (**b**) time-resolves PL of the corresponding samples; (**c**) *V_oc_* vs. different light intensity for 0.75 mol% Co-NiO*_x_* and NiO*_x_*.

**Table 1 nanomaterials-10-00872-t001:** NiO*_x_* processing temperature and the corresponding best efficiency obtained for inverted perovskite solar cells; ITO: Indium tin oxide; PCBM: Phenyl-C61-butyric acid methyl ester; BCP: bathocuproine.

Device Configuration	NiO*_x_* Processing Temperature	Increasing FF (%) by Adding Dopant	PCE (%)	Refs.
Glass/ITO/Co-NiO*_x_*/CH_3_NH_3_PbI_3_/PCBM/BCP/Ag	130 °C	6	16.42	This work
Glass/ITO/Co-NiO*_x_*/CH_3_NH_3_PbI_3_/PCBM/BCP/Ag	130 °C	6	14.5	[23]
Glass/ITO/Co-NiO*_x_*/CH_3_NH_3_PbI_3_/PCBM/PEI/Ag	400 °C	2	18.5	[29]
Glass/FTO/NIR-Co-NiO*_x_*/CH_3_NH_3_PbI_3_/PCBM/PEI/Ag	300 °C	0	17.77	[30]
Glass/ITO/Co-NiO*_x_* (solution-processed)/CH_3_NH_3_PbI_3_/PCBM/C60/Ag	340 °C	10	17.52	[34]

**Table 2 nanomaterials-10-00872-t002:** Photovoltaic performance by *J*/*V* measurements with reverse scan under standard illumination (100 mW cm^−2^) for champion devices and averaged.

Co-doping (mol%)	FF (%)	*J_sc_* (mA cm^−2^)	*V_oc_* (mV)	Best PCE (%)	Average PCE ± s.d. (%)
**0**	70	19.5	968	13.2	11.47 ± 1.08
**0.75**	76	21.5	1005	16.42	14.02 ± 1.3
**1**	75	19.5	920	13.45	11.66 ± 1.1
**1.25**	76	19.75	938	14	11.75 ± 1.3
**2.5**	75	18.7	924	12.9	11.13 ± 0.8
**5**	56	18.7	910	9.5	8.7 ± 0.6

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
