# Peer review of "Structural and Electrical Investigation of Cobalt-Doped NiOx/Perovskite Interface for Efficient Inverted Solar Cells"

_nanomaterials, 2020, doi:10.3390/nano10050872_

Round 1
Reviewer 1 Report
The paper by Rezay Marand and coworkers presents at NiOx with low synthesis temperature as HTM in inverted perovskite solar cells. An approach to improve the conductivity is the metal doping synthesized cobalt-doped NiOx nanoparticles with different concentrations of cobalt (Co) ions and show a champion photoconversion efficiency of 16.42%, with the 0.75% mol of doping.
The paper is well written, should be publish if they clarify the following points:
- Currently there is an article published (https://doi.org/10.1021/acsami.8b01683) which shows almost the same thing, the difference is the process temperature of NiOx and the PCE, could you explain what the novelty of your paper is?
- the fig. 6 show the J-V curve of the different concentrations, its strange to me how with 0.75% Co the PCE jump a lot, could you explant why? and Did you try doped it with less concentration than 0.75%?.
- Also actually is important to show the stability of the devices, add the mppt measurement.
- In the Fig. S3 shows the statistic, plot all the data that you have with the different concentrations, and I saw the average of Co is around 12% and you champion is 16 around 16% there is 4% difference, and in the control is only 2%. Reproducibility with Co is likely poor, explain if that's true.
Author Response
Please, see the answer to the reviewer comments in the attached file.

Reviewer 2 Report
The authors reported on the low temperature deposited Co-doped NiOx HTLs for inverted PSCs and obtained a champion PCE of 16.42 %, with the 0.75 mol % of Co doping. The work carried out by authors can be publishable in Nanomaterials, however the demonstration of the resulted data is really weak and needs further major revision as the conclusions are not well supported by experimental results.
- what is the reason to put “x” in the chemical formula of NiO
- caption of Figure 1; should synthesis of Co-doped NiOx
- Introduction; “This is correlated with the increased fill factor in the solar cells - usually low due to the high temperature deposition (Table 1)” – please provide the corresponding FF values in Table 1
- The authors stated that “The Co-NiOx film consisted of smaller-sized grains than the pure NiOx film and the 0.75 mol% show the most uniform morphology over a large area.” However, from the SEM images shown in Figure 4 it is difficult to see the uniformity. Authors should provide other proof to support the above statement by increasing the magnification of SEM images and/or providing grain size distribution analysis. Moreover, the AFM images and RMS calculation should be provided for all Co-doped NiO films (Figure S4)
- it was stated that “The bigger grain size could be due to better morphology of the Co-doped substrate, as we did not observe difference in the wettability of the perovskite on the different substrates”. However, it is in contradiction with the previous statement where the 0.75 mol% Co-NiO film show the best morphology. The wettability of perovskite drop (contact angle measurements) on all Co-NiO films should be provided and can be helpful to explain the perovskite grain growth.
- again, in another sentence “The uniformity of the inorganic HTM and the big grains of the perovskite, demonstrate the potential of the 0.75 mol% doping for the fabrication of high quality optoelectronic devices.” the bigger grains and thus the reduced grain boundary are observed in 5% mol Co-NiO (Figure 5f) rather than in 0.75% Co-NiO (Figure 5b). Thus, the highest performance in 0.75% Co-NiO based device can not be attributed to the increase of the grain size and improvement of the morphology.
- TRPL data; please provide Table with the calculated fast (τ1) and slow (τ2) decay components for perovskite films deposited on different HTLs.
- Figure 7a and b; spectra for other Co-NiO films should be provided
- as the conductivity and perovskite film quality of 0.75% Co-NiO film is not improved compared to other counterparts, could author explain more detailly what is the reason for the observed significant reduction of average carrier lifetime? Moreover, it was stated that the PL quenching indicates an increase of the non-radiative recombination at the interface. Not decrease?
- please provide equation which was used to fit the dependence of Voc vs light illumination.
Author Response

(The authors gave the same response as above.)

Round 2
Reviewer 1 Report
The paper by Rezay Marand and coworkers presents at NiOx with low synthesis temperature as HTM in inverted perovskite solar cells. An approach to improve the conductivity is the metal doping synthesized cobalt-doped NiOx nanoparticles with different concentrations of cobalt (Co) ions and show a champion photoconversion efficiency of 16.42 %, with the 0.75 mol % of doping.
The paper is well written, should be accepted
Reviewer 2 Report
After reading the revised manuscript and response letter I recommend this work for publication in present form.